# Stability of Stochastic Gradient Method with Momentum for Strongly Convex Loss Functions

## Abstract

While momentum-based methods, in conjunction with the stochastic gradient descent, are widely used when training machine learning models, there is little theoretical understanding on the generalization error of such methods. In practice, the momentum parameter is often chosen in a heuristic fashion with little theoretical guidance. In this work we use the framework of algorithmic stability to provide an upper-bound on the generalization error for the class of strongly convex loss functions, under mild technical assumptions. Our bound decays to zero inversely with the size of the training set, and increases as the momentum parameter is increased. We also develop an upper-bound on the expected true risk, in terms of the number of training steps, the size of the training set, and the momentum parameter.

## 1 Introduction

A fundamental issue for any machine learning algorithm is its ability to *generalize* from the training dataset to the test data. A classical framework used to study the generalization error in machine learning is PAC learning (Vapnik and Chervonenkis, 1971; Valiant, 1984). However, the associated bounds using this approach can be conservative. Recently, the notion of uniform stability, introduced in the seminal work of Bousquet and Elisseeff (Bousquet and Elisseef, 2002), is leveraged to analyze the generalization error of the stochastic gradient method (SGM) (Hardt et al., 2016). The result in (Hardt et al., 2016) is a substantial step forward, since SGM is widely used in many practical systems. This method is scalable, robust, and widely adopted in a broad range of problems.

To accelerate the convergence of SGM, a momentum term is often added in the iterative update of the stochastic gradient (Goodfellow et al., 2016). This approach has a long history, with proven benefits in various settings. The heavy-ball momentum method was first introduced by Polyak (Polyak, 1964), where a weighted version of the previous update is added to the current gradient update. Polyak motivated his method by its resemblance to a heavy ball moving in a potential well defined by the objective function. Momentum methods have been used to accelerate the back-propagation algorithm when training neural networks (Rumelhart et al., 1986). Intuitively, adding momentum accelerates convergence by circumventing sharp curvatures and long ravines of the sub-level sets of the objective function (Wilson et al., 2018). For example, Ochs et al. has presented an illustrative example to show that the momentum can potentially avoid local minima (Ochs et al., 2015). Nesterov has proposed an accelerated gradient method, which converges as $O(1/k^2)$ where $k$ is the number of iterations (Nesterov, 1983). However, the Netstrov momentum does not seem to improve the rate of convergence for stochastic gradient (Goodfellow et al., 2016, Section 8.3.3). In this work, we focus on the heavy-ball momentum.

Although momentum methods are well known to improve the convergence in SGM, their effect on the generalization error is not well understood. In this work, we first build upon the framework in (Hardt et al., 2016) to obtain a bound on the generalization error of SGM with momentum (SGMM) for the case of strongly convex loss functions. Our bound is independent of the number of training iterations and decreases inversely with the size of the training set. Secondly, we develop an upper-bound on the optimization error, which quantifies the gap between the empirical risk of SGMM and the global optimum. Our bound can be made arbitrarily small by choosing sufficiently many iterations and a sufficiently small learning rate. Finally, we establish an upper-bound on the

expected true risk of SGMM as a function of various problem parameters. We note that the class of strongly convex loss functions appears in several important machine learning problems, including linear and logistic regression with a weight decay regularization term.

**Other related works:** convergence analysis of first order methods with momentum is studied in (Nesterov, 1983; Ochs et al., 2014; Su et al., 2014; Ghadimi et al., 2015; Lessard et al., 2016; Yang et al., 2016; Loizou and Richtárik, 2018; Gadat et al., 2016). Most of these works consider the deterministic setting for gradient update. Only a few works have analyzed the stochastic setting (Yang et al., 2016; Loizou and Richtárik, 2018; Gadat et al., 2016). Our convergence analysis results are not directly comparable with these works due to their different assumptions regarding the properties of loss functions. In particular, we analyze the convergence of SGMM for a smooth and strongly convex loss function as in (Hardt et al., 2016), which is new.

First-order methods with noisy gradient are studied in (Kidambi et al., 2018) and references therein. In (Kidambi et al., 2018), the authors show that there exists linear regression problems for which SGM outperforms SGMM in terms of convergence.

Our main focus in this work is on the generalization, and hence true risk, of SGMM. We are aware of only one similar work in this regard, which provides stability bounds for quadratic loss functions (Chen et al., 2018). In this paper, we obtain stability bounds for the general case of strongly convex loss functions. In addition, unlike (Chen et al., 2018), our results show that machine learning models can be trained for multiple epochs of SGMM with bounded generalization errors.

**Notation:** We use $\mathbb{E}[\cdot]$ to denote the expectation and $\|\cdot\|$ to represent the Euclidean norm of a vector. We use lower-case bold font to denote vectors. We use sans-serif font to denote random quantities. Sets and scalars are represented by calligraphic and standard fonts, respectively.

## 2 GENERALIZATION ERROR AND STABILITY

We consider a general supervised learning problem, where $\mathcal{S} = \{\mathbf{z}_1, \cdots, \mathbf{z}_n\}$ denotes the set of samples of size $n$ drawn i.i.d. from some space $\mathcal{Z}$ with an unknown distribution $D$. We assume a learning model described by parameter vector $\mathbf{w}$. Let $f(\mathbf{w}; \mathbf{z})$ denote the loss of the model described by parameter $\mathbf{w}$ on example $\mathbf{z} \in \mathcal{Z}$. Our ultimate goal is to minimize the true or population risk:

$$R(\mathbf{w}) \triangleq \mathbb{E}_{\mathbf{z} \sim D} f(\mathbf{w}; \mathbf{z}). \tag{1}$$

Since the distribution $D$ is unknown, we replace the objective by the empirical risk, *i.e.,*

$$R_S(\mathbf{w}) \triangleq \frac{1}{n} \sum_{i=1}^{n} f(\mathbf{w}; \mathbf{z}_i). \tag{2}$$

We assume $\mathbf{w} = A(\mathcal{S})$ for a potentially randomized algorithm $A(\cdot)$. In order to find an upper-bound on the true risk, we consider the generalization error, which is the expected difference of empirical and true risk:

$$\epsilon_g \triangleq \mathbb{E}_{\mathcal{S},A}[R(A(\mathcal{S})) - R_S(A(\mathcal{S}))]. \tag{3}$$

Finally, to upper bound $\epsilon_g$, we consider uniform stability:

**Definition 1** *Let $\mathcal{S}$ and $\mathcal{S}'$ denote two data sets from space $\mathcal{Z}^n$ such that $\mathcal{S}$ and $\mathcal{S}'$ differ in at most one example. Algorithm $A$ is $\epsilon_s$-uniformly stable if for all data sets $\mathcal{S}, \mathcal{S}'$, we have*

$$\sup_{\mathbf{z}} \mathbb{E}_A[f(A(\mathcal{S}); \mathbf{z}) - f(A(\mathcal{S}'); \mathbf{z})] \le \epsilon_s. \tag{4}$$

It is shown in (Hardt et al., 2016) that uniform stability implies generalization in expectation:

**Theorem 1** *(Hardt et al., 2016) If $A$ is an $\epsilon_s$-uniformly stable algorithm, then the generalization error of $A$ is upper-bounded by $\epsilon_s$.*

Theorem 1 shows that it is enough to control the uniform stability of an algorithm to upper bound the generalization error.

## 2.1 Assumptions on the loss function

In our analysis, we will assume that the loss function satisfies the following properties.

**Definition 2** *A function $f : \Psi \to \mathbb{R}$ is L-Lipschitz if for all $\mathbf{u}$, $\mathbf{v} \in \Psi$ we have $|f(\mathbf{u}) - f(\mathbf{v})| \leq L\|\mathbf{u} - \mathbf{v}\|$.*

**Definition 3** *A function $f : \Psi \to \mathbb{R}$ is $\beta$-smooth if for all $\mathbf{u}$, $\mathbf{v} \in \Psi$ we have $\|\nabla f(\mathbf{u}) - \nabla f(\mathbf{v})\| \leq \beta\|\mathbf{u} - \mathbf{v}\|$.*

**Definition 4** *A function $f : \Psi \to \mathbb{R}$ is $\gamma$-strongly convex if for all $\mathbf{u}$, $\mathbf{v} \in \Psi$ we have $f(\mathbf{u}) \geq f(\mathbf{v}) + \nabla f(\mathbf{v})^T(\mathbf{u} - \mathbf{v}) + \frac{\gamma}{2}\|\mathbf{u} - \mathbf{v}\|^2$.*

We assume that the parameter space $\Omega$ is a convex set. Furthermore, for the loss function to be $L$-Lipschitz and and strongly convex, we further assume that $\Omega$ is compact. Since $\Omega$ is compact, the SGMM update requires projection.

## 2.2 Stochastic gradient method with momentum

The update rule for projected SGMM is given by:

$$\mathbf{w}_{t+1} = \mathbf{P}\Big(\mathbf{w}_t + \mu(\mathbf{w}_t - \mathbf{w}_{t-1}) - \alpha\nabla_{\mathbf{w}}f(\mathbf{w}_t; \mathbf{z}_{i_t})\Big) \tag{5}$$

where $\mathbf{P}$ denotes the Euclidean projection onto $\Omega$, $\alpha > 0$ is the learning rate[1], $\mu > 0$ is the momentum parameter, $i_t$ is a randomly selected index, and $f(\mathbf{w}_t; \mathbf{z}_{i_t})$ is the loss evaluated on sample $\mathbf{z}_{i_t}$. In SGMM, we run the update (5) iteratively for $T$ steps and let $\mathbf{w}_T$ denote the final output. Note that there are two typical approaches to select $i_t$. The first approach is to select $i_t \in \{1, \cdots, n\}$ uniformly at random at each iteration. The second approach is to permutate $\{1, \cdots, n\}$ randomly once and then select the examples repeatedly in a cyclic manner. Our results are valid for both approaches. The key quantity of interest in this paper is the generalization error for SGMM given by:

$$\epsilon_g = \mathbb{E}_{\mathcal{S},A}[R(\mathbf{w}_T) - R_S(\mathbf{w}_T)] = \mathbb{E}_{\mathcal{S},i_0,\cdots,i_{T-1}}[R(\mathbf{w}_T) - R_S(\mathbf{w}_T)]$$

since the randomness in $A$ arises from the choice of $i_0, \cdots, i_{T-1}$.

# 3 Main results

In the following, we assume that the loss function $f(\cdot; \mathbf{z})$ is $\beta$-smooth, $L$-Lipschitz, and $\gamma$-strongly convex for all $\mathbf{z}$.

**Theorem 2 (Stability bound)** *Suppose that the SGMM update (5) is executed for $T$ steps with constant learning rate $\alpha$ and momentum $\mu$. Provided that $\frac{\alpha\beta\gamma}{\beta+\gamma} - \frac{1}{2} \leq \mu < \frac{\alpha\beta\gamma}{3(\beta+\gamma)}$ and $\alpha \leq \frac{2}{\beta+\gamma}$, SGMM satisfies $\epsilon_s$-uniform stability where*

$$\epsilon_s \leq \frac{2\alpha L^2(\beta + \gamma)}{n\big(\alpha\beta\gamma - 3\mu(\beta + \gamma)\big)}. \tag{6}$$

The result in Theorem 2 implies that the stability bound decreases inversely with the size of the training set. It increases as the momentum parameter $\mu$ increases. These properties are also verified in our experimental evaluation.

**Theorem 3 (Convergence bound)** *Suppose that the SGMM update (5) is executed for $T$ steps with constant learning rate $\alpha$ and momentum $\mu$. Then we have*

$$\mathbb{E}_{\mathcal{S},A}[R_S(\hat{\mathbf{w}}_T) - R_S(\mathbf{w}_S^*)] \leq \frac{\mu W_0}{(1-\mu)T} + \frac{(1-\mu)W_1}{2\alpha T} - \frac{\gamma W_2}{2} - \frac{\mu\gamma W_3}{2(1-\mu)} + \frac{\alpha L^2}{2(1-\mu)} \tag{7}$$

---

[1]In the following, we assume $\alpha$ is constant over time. In practice, time-decaying $\alpha$ is used in some applications. Please note that our results hold for time-decaying $\alpha$.

where $\hat{\mathbf{w}}_T$ denotes the average of $T$ steps of the algorithm, i.e., $\hat{\mathbf{w}}_T = \frac{1}{T+1} \sum_{t=0}^{T} \mathbf{w}_t$, $R_S(\mathbf{w}) = \frac{1}{n} \sum_{i=1}^{n} f(\mathbf{w}; \mathbf{z}_i)$, $\mathbf{w}_S^* = \arg\min_{\mathbf{w}} R_S(\mathbf{w})$, $W_0 = \mathbb{E}_{\mathcal{S},A}[R_S(\mathbf{w}_0) - R_S(\mathbf{w}_T)]$, $W_1 = \mathbb{E}_{\mathcal{S},A}[\|\mathbf{w}_0 - \mathbf{w}_S^*\|^2]$, $W_2 = \mathbb{E}_{\mathcal{S},A}[\|\hat{\mathbf{w}}_T - \mathbf{w}_S^*\|^2]$, and $W_3 = \frac{1}{T+1} \sum_{t=0}^{T} \mathbb{E}_{\mathcal{S},A}[\|\mathbf{w}_t - \mathbf{w}_{t-1}\|^2]$.

Theorem 3 bounds the optimization error, *i.e.,* the expected difference between the empirical risk achieved by SGMM and the global minimum. Upon setting $\mu = 0$ and $\gamma = 0$ in (7), we can recover the classical bound on optimization error for SGM (Nemirovski and Yudin., 1983), (Hardt et al., 2016, Theorem 5.2). The first two terms in (7) vanish as $T$ increases. The terms with negative sign improve the convergence due to the strongly convexity. The last term depends on the learning rate, $\alpha$, the momentum parameter $\mu$, and the Lipschitz constant $L$. This term can be controlled by selecting $\alpha$ sufficiently small.

**Proposition 1 (Upper-bound on true risk)** *Suppose that the SGMM update* (5) *is executed for $T$ steps with constant learning rate $\alpha$ and momentum $\mu$, satisfying the conditions in Theorem 2 and $\mu(\beta + \gamma) \ll \alpha\beta\gamma$. Then, setting $\alpha = \frac{1-\mu}{L}\sqrt{\frac{W_1}{T}}$, we have:*

$$\mathbb{E}_{\mathcal{S},A}[R(\hat{\mathbf{w}}_T)] \leq \mathbb{E}_{\mathcal{S},A}[R_S(\mathbf{w}_S^*)] + \frac{\mu W_0}{(1-\mu)T} + L\sqrt{\frac{W_1}{T}} - \frac{\gamma W_2}{2} - \frac{\mu\gamma W_3}{2(1-\mu)} + \frac{2L^2(\beta+\gamma)}{n\beta\gamma C}$$

(8)

*where $C \triangleq 1 - \frac{3\mu L(\beta+\gamma)\sqrt{T}}{(1-\mu)\beta\gamma\sqrt{W_1}}$ and $\hat{\mathbf{w}}_T$ as well as the constants $W_0, \cdots, W_3$ are defined in Theorem 3.*

Proposition 1 provides a bound on the expected true risk of SGMM in terms of the global minimum of the empirical risk. The bound in (8) is obtained by combining Theorem 2 and Theorem 3 and minimizing the expression over $\alpha$. The choice of $\alpha$ simplifies considerably when $\mu$ is sufficiently small, as stated in Proposition 1. Due to the page constraint, the proof of this result is provided in the supplementary material. Note that the first two terms in (8) vanish as $T$ increases. The last term in (8) vanishes as the number of samples $n$ increases.

## 4 PROOF OF THEOREM 2 (STABILITY BOUND)

Following (Hardt et al., 2016), we track the divergence of two different iterative sequences of update rules with the same starting point. However, our analysis is more involved as the presence of momentum term requires a more careful bound on the iterative expressions.

To keep the notation uncluttered, we first consider SGMM without projection and defer the discussion of projection to the end of this proof. Let $\mathcal{S} = \{\mathbf{z}_1, \cdots, \mathbf{z}_n\}$ and $\mathcal{S}' = \{\mathbf{z}'_1, \cdots, \mathbf{z}'_n\}$ be two samples of size $n$ that differ in at most one example. Let $\mathbf{w}_T$ and $\mathbf{w}'_T$ denote the outputs of SGMM on $\mathcal{S}$ and $\mathcal{S}'$, respectively. We consider the updates $\mathbf{w}_{t+1} = G_t(\mathbf{w}_t) + \mu(\mathbf{w}_t - \mathbf{w}_{t-1})$ and $\mathbf{w}'_{t+1} = G'_t(\mathbf{w}'_t) + \mu(\mathbf{w}'_t - \mathbf{w}'_{t-1})$ with $G_t(\mathbf{w}_t) = \mathbf{w}_t - \alpha\nabla_{\mathbf{w}} f(\mathbf{w}_t; \mathbf{z}_{i_t})$ and $G'_t(\mathbf{w}'_t) = \mathbf{w}'_t - \alpha\nabla_{\mathbf{w}} f(\mathbf{w}'_t; \mathbf{z}'_{i_t})$, respectively, for $t = 1, \cdots, T$. We denote $\delta_t \triangleq \|\mathbf{w}_t - \mathbf{w}'_t\|$. Suppose $\mathbf{w}_0 = \mathbf{w}'_0$, *i.e.,* $\delta_0 = 0$. We first establish an upper-bound on $\mathbb{E}_A[\delta_{t+1}]$ in terms of $\mathbb{E}_A[\delta_t]$ and $\mathbb{E}_A[\delta_{t-1}]$ in the following lemma, whose proof is provided in the supplementary document.

**Lemma 1** *Provided that $\alpha \leq \frac{2}{\beta+\gamma}$, an upper-bound on $\mathbb{E}_A[\delta_{t+1}]$ is given by*

$$\mathbb{E}_A[\delta_{t+1}] \leq \left(1 + \mu - \frac{\alpha\beta\gamma}{\beta+\gamma}\right)\mathbb{E}_A[\delta_t] + \mu\mathbb{E}_A[\delta_{t-1}] + \frac{2\alpha L}{n}.$$

(9)

Using the result of Lemma 1, in the following, we develop an upper bound on $\mathbb{E}_A[\delta_T]$. Let us consider the recursion

$$\mathbb{E}_A[\tilde{\delta}_{t+1}] = \left(1 + \mu - \frac{\alpha\beta\gamma}{\beta+\gamma}\right)\mathbb{E}_A[\tilde{\delta}_t] + \mu\mathbb{E}_A[\tilde{\delta}_{t-1}] + \frac{2\alpha L}{n}$$

(10)

with $\tilde{\delta}_0 = \delta_0 = 0$. Upon inspecting (10) it is clear that

$$\mathbb{E}_A[\tilde{\delta}_t] \geq \left(1 + \mu - \frac{\alpha\beta\gamma}{\beta+\gamma}\right)\mathbb{E}_A[\tilde{\delta}_{t-1}], \qquad \forall t \geq 1,$$

(11)

as we simply drop the remainder of positive terms. Substituting (11) into (10), we have

$$\mathbb{E}_A[\tilde{\delta}_{t+1}] \leq \left(1 + \mu + \frac{\mu}{1 + \mu - \frac{\alpha\beta\gamma}{\beta+\gamma}} - \frac{\alpha\beta\gamma}{\beta+\gamma}\right)\mathbb{E}_A[\tilde{\delta}_t] + \frac{2\alpha L}{n}$$

$$\leq \left(1 + 3\mu - \frac{\alpha\beta\gamma}{\beta+\gamma}\right)\mathbb{E}_A[\tilde{\delta}_t] + \frac{2\alpha L}{n} \tag{12}$$

where the second inequality holds due to $\mu \geq \frac{\alpha\beta\gamma}{\beta+\gamma} - \frac{1}{2}$.

Noting that $\mathbb{E}_A[\tilde{\delta}_t] \geq \mathbb{E}_A[\delta_t]$ for all $t$ including $T$, we have

$$\mathbb{E}_A[\delta_T] \leq \frac{2\alpha L}{n}\sum_{t=1}^{T}\left(1 + 3\mu - \frac{\alpha\beta\gamma}{\beta+\gamma}\right)^t \leq \frac{2\alpha L(\beta+\gamma)}{n\big(\alpha\beta\gamma - 3\mu(\beta+\gamma)\big)} \tag{13}$$

where the second expression holds since $0 \leq \mu < \frac{\alpha\beta\gamma}{3(\beta+\gamma)}$ is assumed.

Applying the $L$-Lipschitz property on $f(\cdot, \mathbf{z})$, it follows that

$$\mathbb{E}_A[|f(\mathbf{w}_T; \mathbf{z}) - f(\mathbf{w}'_T; \mathbf{z})|] \leq L\mathbb{E}_A[\delta_T] \leq \frac{2\alpha L^2(\beta+\gamma)}{n\big(\alpha\beta\gamma - 3\mu(\beta+\gamma)\big)}. \tag{14}$$

Since this bound holds for all $\mathcal{S}, \mathcal{S}'$ and $\mathbf{z}$, we obtain an upper-bound on the uniform stability and the proof is complete. Our stability bound in Theorem 2 holds for the projected SGMM update (5) because Euclidean projection does not increase the distance between projected points (the argument is essentially analogous to (Hardt et al., 2016, Lemma 4.6)). In particular, note that Lemma 1 holds for the projected SGMM.

## 5 PROOF OF THEOREM 3 (CONVERGENCE BOUND)

Again, we first consider SGMM without projection and discuss the extension to projection at the end of this proof. Our proof is inspired by the convergence analysis in (Yang et al., 2016; Ghadimi et al., 2015) for a convex loss function with bounded variance and time-decaying learning rate. Different from these works, we analyze the convergence of SGMM for a smooth and strongly convex loss function with constant learning rate. To facilitate the convergence analysis, we define:

$$\mathbf{p}_t \triangleq \frac{\mu}{1-\mu}(\mathbf{w}_t - \mathbf{w}_{t-1}) \tag{15}$$

with $\mathbf{p}_0 = 0$. Substituting into the SGMM update, the parameter recursion is given by

$$\mathbf{w}_{t+1} + \mathbf{p}_{t+1} = \mathbf{w}_t + \mathbf{p}_t - \frac{\alpha}{1-\mu}\nabla_{\mathbf{w}}f(\mathbf{w}_t; \mathbf{z}_{i_t}) \tag{16}$$

It follows that

$$\|\mathbf{w}_{t+1} + \mathbf{p}_{t+1} - \mathbf{w}\|^2 = \|\mathbf{w}_t + \mathbf{p}_t - \mathbf{w}\|^2 + \left(\frac{\alpha}{1-\mu}\right)^2\|\nabla_{\mathbf{w}}f(\mathbf{w}_t; \mathbf{z}_{i_t})\|^2$$

$$- \frac{2\alpha}{1-\mu}(\mathbf{w}_t + \mathbf{p}_t - \mathbf{w})^T\nabla_{\mathbf{w}}f(\mathbf{w}_t; \mathbf{z}_{i_t}). \tag{17}$$

Substituting $\mathbf{p}_t$ (15) into (17), the recursion (16) can be written as

$$\|\mathbf{w}_{t+1} + \mathbf{p}_{t+1} - \mathbf{w}\|^2 = \|\mathbf{w}_t + \mathbf{p}_t - \mathbf{w}\|^2 + \left(\frac{\alpha}{1-\mu}\right)^2\|\nabla_{\mathbf{w}}f(\mathbf{w}_t; \mathbf{z}_{i_t})\|^2$$

$$- \frac{2\alpha\mu}{(1-\mu)^2}(\mathbf{w}_t - \mathbf{w}_{t-1})^T\nabla_{\mathbf{w}}f(\mathbf{w}_t; \mathbf{z}_{i_t}) - \frac{2\alpha}{1-\mu}(\mathbf{w}_t - \mathbf{w})^T\nabla_{\mathbf{w}}f(\mathbf{w}_t; \mathbf{z}_{i_t}). \tag{18}$$

Upon taking the expectation with respect to $i_t$ in (18) we have

$$\mathbb{E}_{i_t}\|\mathbf{w}_{t+1} + \mathbf{p}_{t+1} - \mathbf{w}\|^2 \leq \|\mathbf{w}_t + \mathbf{p}_t - \mathbf{w}\|^2 + \left(\frac{\alpha}{1-\mu}\right)^2 L^2 - \frac{2\alpha\mu}{(1-\mu)^2}(\mathbf{w}_t - \mathbf{w}_{t-1})^T\nabla_{\mathbf{w}}R_S(\mathbf{w}_t)$$

$$- \frac{2\alpha}{1-\mu}(\mathbf{w}_t - \mathbf{w})^T\nabla_{\mathbf{w}}R_S(\mathbf{w}_t) \tag{19}$$

where we use the fact that $\|\nabla_{\mathbf{w}} f(\mathbf{w}_t; \mathbf{z}_{i_t})\| \leq L$, due to $L$-Lipschitz, and that $\mathbb{E}_{i_t}[\nabla_{\mathbf{w}} f(\mathbf{w}_t; \mathbf{z}_{i_t})] = \nabla_{\mathbf{w}} R_S(\mathbf{w}_t)$. Furthermore, since $R_S(\cdot)$ is a $\gamma$-strongly convex function, for all $\mathbf{w}_t$ and $\mathbf{w}_{t-1}$, we have

$$(\mathbf{w}_t - \mathbf{w})^T \nabla_{\mathbf{w}} R_S(\mathbf{w}_t) \geq R_S(\mathbf{w}_t) - R_S(\mathbf{w}) + \frac{\gamma}{2}\|\mathbf{w}_t - \mathbf{w}\|^2,$$

$$(\mathbf{w}_t - \mathbf{w}_{t-1})^T \nabla_{\mathbf{w}} R_S(\mathbf{w}_t) \geq R_S(\mathbf{w}_t) - R_S(\mathbf{w}_{t-1}) + \frac{\gamma}{2}\|\mathbf{w}_t - \mathbf{w}_{t-1}\|^2. \tag{20}$$

Substituting (20) in (19), we have

$$\mathbb{E}_{i_t}[\|\mathbf{w}_{t+1} + \mathbf{p}_{t+1} - \mathbf{w}\|^2] \leq \|\mathbf{w}_t + \mathbf{p}_t - \mathbf{w}\|^2 - \frac{\alpha\gamma}{1-\mu}\|\mathbf{w}_t - \mathbf{w}\|^2 - \frac{2\alpha\mu}{(1-\mu)^2}\big(R_S(\mathbf{w}_t) - R_S(\mathbf{w}_{t-1})\big)$$

$$- \frac{2\alpha}{1-\mu}\big(R_S(\mathbf{w}_t) - R_S(\mathbf{w})\big) + \frac{\alpha^2 L^2}{(1-\mu)^2} - \frac{\alpha\mu\gamma}{(1-\mu)^2}\|\mathbf{w}_t - \mathbf{w}_{t-1}\|^2. \tag{21}$$

Taking expectation over $i_0, \cdots, i_t$ for a given $\mathcal{S}$, summing (21) for $t = 0, \cdots, T$, and rearranging terms, we have

$$\frac{2\alpha}{1-\mu}\sum_{t=0}^{T}\mathbb{E}_A[R_S(\mathbf{w}_t) - R_S(\mathbf{w})] \leq \frac{2\alpha\mu}{(1-\mu)^2}\mathbb{E}_A[R_S(\mathbf{w}_0) - R_S(\mathbf{w}_T)] - \frac{\alpha\gamma}{1-\mu}\sum_{t=0}^{T}\mathbb{E}_A[\|\mathbf{w}_t - \mathbf{w}\|^2]$$

$$- \frac{\alpha\mu\gamma}{(1-\mu)^2}\sum_{t=0}^{T}\mathbb{E}_A[\|\mathbf{w}_t - \mathbf{w}_{t-1}\|^2] + \mathbb{E}_A[\|\mathbf{w}_0 - \mathbf{w}\|^2] + \frac{\alpha^2 L^2 (T+1)}{(1-\mu)^2}. \tag{22}$$

Since $\|\cdot\|$ is a convex function, for all $\mathbf{w}_T$ and $\mathbf{w}$, we have

$$\|\hat{\mathbf{w}}_T - \mathbf{w}\|^2 \leq \frac{1}{T+1}\sum_{t=0}^{T}\|\mathbf{w}_t - \mathbf{w}\|^2. \tag{23}$$

Furthermore, due to convexity of $R_S(\cdot)$, we have

$$R_S(\hat{\mathbf{w}}_T) - R_S(\mathbf{w}) \leq \frac{1}{T+1}\sum_{t=0}^{T}\big(R_S(\mathbf{w}_t) - R_S(\mathbf{w})\big). \tag{24}$$

Taking expectation over $\mathcal{S}$, applying inequalities (23) and (24) into (22), and substituting $\mathbf{w} = \mathbf{w}_S^*$, we obtain (7) and the proof is complete.

Our convergence bound in Theorem 3 can be extended to projected SGMM (5). Let use denote $\mathbf{y}_{t+1} \overset{\Delta}{=} \mathbf{w}_t + \mu(\mathbf{w}_t - \mathbf{w}_{t-1}) - \alpha\nabla_{\mathbf{w}} f(\mathbf{w}_t; \mathbf{z}_{i_t})$. Then, for any feasible $\mathbf{w} \in \Omega$, (17) holds for $\mathbf{y}_{t+1}$, i.e.,

$$\|\mathbf{y}_{t+1} + \frac{\mu}{1-\mu}(\mathbf{y}_{t+1} - \mathbf{w}_t) - \mathbf{w}\|^2 = \|\mathbf{w}_t + \mathbf{p}_t - \mathbf{w}\|^2 + \big(\frac{\alpha}{1-\mu}\big)^2\|\nabla_{\mathbf{w}} f(\mathbf{w}_t; \mathbf{z}_{i_t})\|^2$$

$$- \frac{2\alpha}{1-\mu}(\mathbf{w}_t + \mathbf{p}_t - \mathbf{w})^T \nabla_{\mathbf{w}} f(\mathbf{w}_t; \mathbf{z}_{i_t}). \tag{25}$$

Note that the LHS of (25) can be written as

$$\|\mathbf{y}_{t+1} + \frac{\mu}{1-\mu}(\mathbf{y}_{t+1} - \mathbf{w}_t) - \mathbf{w}\|^2 = \frac{1}{1-\mu}\|\mathbf{y}_{t+1} - \big(\mu\mathbf{w}_t + (1-\mu)\mathbf{w}\big)\|.$$

We note that $\mu\mathbf{w}_t + (1-\mu)\mathbf{w} \in \Omega$ for any $\mathbf{w} \in \Omega$ and $\mathbf{w}_t \in \Omega$ since $\Omega$ is convex.

Now in projected SGMM, we have

$$\|\mathbf{w}_{t+1} - \big(\mu\mathbf{w}_t + (1-\mu)\mathbf{w}\big)\|^2 = \|\mathbf{P}(\mathbf{y}_{t+1}) - \big(\mu\mathbf{w}_t + (1-\mu)\mathbf{w}\big)\|^2$$

$$\leq \|\mathbf{y}_{t+1} - \big(\mu\mathbf{w}_t + (1-\mu)\mathbf{w}\big)\|^2 \tag{26}$$

since projection a point onto $\Omega$ moves it closer to any point in $\Omega$. This shows inequality (19) holds, and the convergence results do not change.

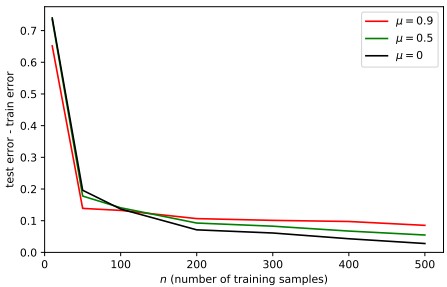 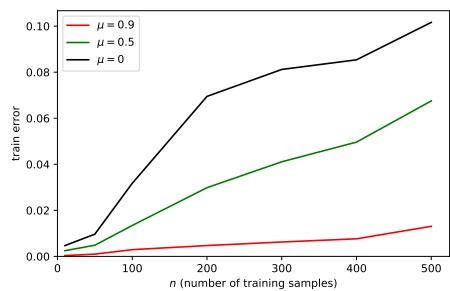

(a) Generalization error with respect to cross entropy    (b) Training error (cross entropy)

Figure 1: Generalization performance (cross entropy) of logistic regression for notMNIST dataset with $T = 1000$ iterations and minibatch size 10.

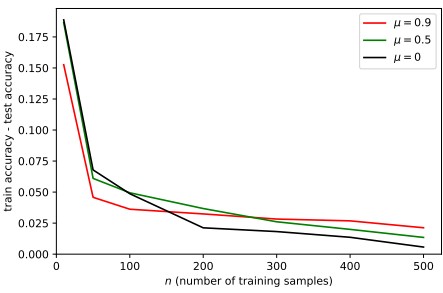 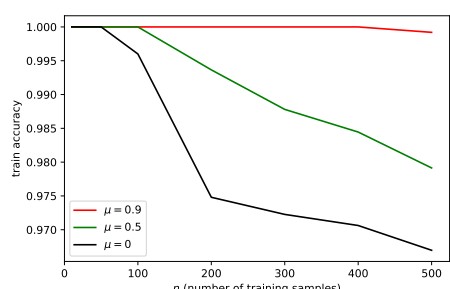

(a) Generalization error with respect to classification accuracy

(b) Training accuracy

Figure 2: Generalization performance (classification accuracy) of logistic regression for notMNIST dataset with $T = 1000$ iterations and minibatch size 10.

## 6 EXPERIMENTAL EVALUATION

In this section, we validate the insights obtained in our theoretical results in experimental evaluation. Our main goal is to study how adding momentum affects the convergence and generalization of SGM. We study the performance of SGMM when applied to the notMINIST dataset. Please note that similar results are provided for the MNIST dataset in the supplementary document. We train a logistic regression model with the weight decay regularization using SGMM for binary classification on the two-class notMNIST dataset that contains the images from letter classes "C" and "J", which leads to a smooth and strongly convex loss function. We set the learning rate $\alpha = 0.01$. The weight decay coefficient and the minibatch size are set to 0.001 and 10, respectively. We use 100 SGMM realizations to evaluate the average performance. We compare the training and generalization performance of SGM without momentum with that of SGMM under $\mu = 0.5$ and $\mu = 0.9$, which are common momentum values used in practice (Goodfellow et al., 2016, Section 8.3.2).

The generalization error (with respect to cross entropy) and training error versus the number of training samples, $n$, under SGMM with fixed $T = 1000$ iterations are shown in Figures 1a and 1b, respectively, for $\mu = 0, 0.5, 0.9$. In Figures 2a and 2b, we plot the generalization error (with respect to classification accuracy) and the training accuracy as a function of the number of training samples for the same dataset. First, we observe that the generalization error (with respect to both cross entropy and classification accuracy) decreases as $n$ increases for all values of $\mu$, which is suggested by our stability upper-bound in Theorem 2. In addition, for sufficiently large $n$, we observe that the generalization error increases with $\mu$, consistent with Theorem 2. On the other hand, the training error increases as $n$ increases, which is expected. We can observe that adding momentum reduces training error as it improves the convergence rate. The training accuracy also improves by adding momentum as illustrated in Fig. 2b.

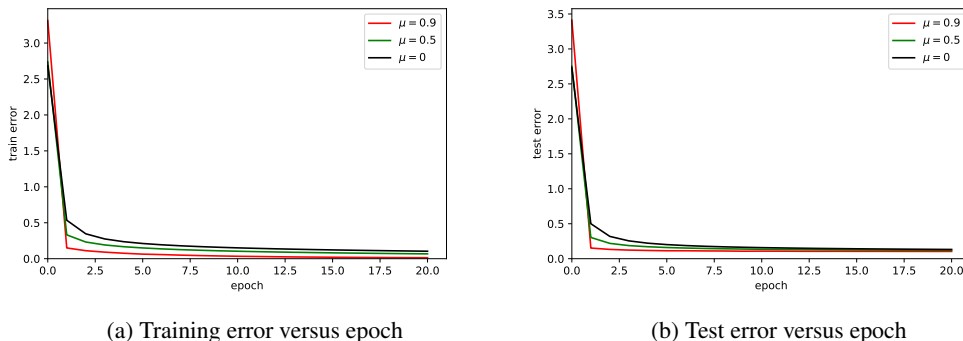

(a) Training error versus epoch

(b) Test error versus epoch

Figure 3: Training and test error of logistic regression (cross entropy loss) for notMNIST dataset with $n = 500$ training samples and minibatch size 10.

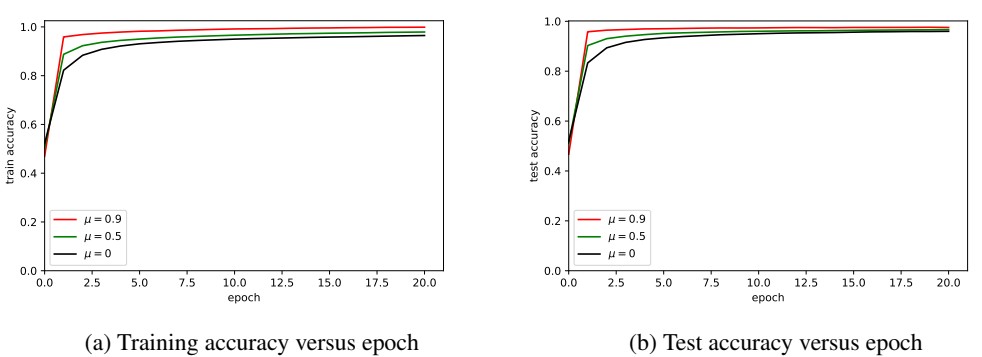

(a) Training accuracy versus epoch

(b) Test accuracy versus epoch

Figure 4: Training and test accuracy of logistic regression for notMNIST dataset with $n = 500$ training samples and minibatch size 10.

In order to study the optimization error of SGMM, we show the training error and test error versus the number of epochs, under SGMM trained with $n = 500$ samples in Figures 3a and 3b, respectively. We plot the classification accuracy for training and test datasets in Figures 4a and 4b, respectively. We observe that the training error decreases as the number of epochs increases for all values of $\mu$, which is consistent with the convergence analysis in Theorem 3. Furthermore, as expected, we see that adding momentum improves the training error and accuracy. However, as the number of epochs increases, we note that the benefit of momentum on the test error and accuracy becomes negligible. This happens because adding momentum also results in a higher generalization error thus penalizing the gain in training error.

## 7 CONCLUSIONS

We study the generalization error and convergence of SGMM for the class of strongly convex loss functions, under mild technical conditions. We establish an upper-bound on the generalization error, which decreases with the size of the training set, and increases as the momentum parameter is increased. Secondly, we analyze the convergence of SGMM during training, by establishing an upper-bound on the gap between the empirical risk of SGMM and the global minimum. Our proposed bound reduces to a classical bound on the optimization error of SGM (Nemirovski and Yudin., 1983) for convex functions, when the momentum parameter is set to zero. Finally, we establish an upper-bound on the expected difference between the true risk of SGMM and the global minimum of the empirical risk, and illustrate how it scales with the number of training steps and the size of the training set. Although our results are established for the case when the learning rate is constant, they can be easily extended to the case when the learning rate decreases with the number of iterations. We also present experimental evaluations on the notMNIST dataset and show that the numerical plots are consistent with our theoretical bounds on the generalization error and the convergence gap.

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
