# OpenReview forum: "Stability of Stochastic Gradient Method with Momentum for Strongly Convex Loss Functions"
_ICLR.cc/2019/Conference_

### Official Review · AnonReviewer2 · 2018-10-27
**Stability of Stochastic Gradient Method with Momentum for Strongly Convex Loss Functions**

**Rating:** 4
**Confidence:** 4

**Review:**

Comments:

The author(s) provide stability and generalization bounds for SGD with momentum for strongly convex, smooth, and Lipschitz losses.

This paper basically follows and extends the results from (Hardt, Recht, and Singer, 2016). Section 2 is quite identical but without mentioning the overlap from Section 2 in (Hardt et al, 2016). The analysis closely follows the approach from there.

The proof of Theorem 2 has some issues. The set of assumptions (smooth, Lipschitz and strongly convex) is not valid on the whole set R^d, for example quadratic function. In this case, your Lipschitz constant L would be arbitrarily large and could be damaged your theoretical result. To consider projected step is true, but the proof without projection (and then explaining in the end) should have troubles.

From the theoretical results, it is not clear that momentum parameter affects positively or negatively. In Theorem 3, what is the advantage of this convergence compared to SGD? It seems that it is not better than SGD. Moreover, if \mu = 0 and \gamma > 0, it seems not able to recover the linear convergence to neighborhood of SGD. Please also notice that, in this situation, L also could be large.

The topic could be interesting but the contributions are very incremental. At the current state, I do not support the publications of this paper.

---

> ### Author Response · Authors · 2018-11-27
> **Response to AnonReviewer2 Comments**
>
> R2C1: We believe that our results are substantial and important. Our analysis involves some subtle but important steps in dealing with the momentum term in the recursion in Section 4. This method was not conceived in prior attempts on this problem. We reproduce the following statement from [Hardt et al., Section 7]: ``One very important technique that we did not discuss is momentum. However, it is not clear that momentum adds stability. It is possible that momentum speeds up training but adversely impacts generalization.'' Our work is the first successful attempt that establishes that SGMM generalizes, for the practically important class of strongly convex loss functions.
> ------------------------------------------------------------------------------------------------------------------------------------------------------------------------
> R2C2: Please note that we first discuss the proofs without projection to keep the notation uncluttered. We then explain how the proofs can be modified to accommodate projection. We believe this approach is technically sound, and it helps the readers to better understand the insights in our proofs. We respectfully request that the reviewer point out any specific issue in our proofs such that we can fix it.
> ------------------------------------------------------------------------------------------------------------------------------------------------------------------------
> R2C3: To the best of our understanding, linear convergence happens under a very stringent condition: $\Pr\{\nabla f_i(x^*)=0\}=1$, \ie $x^*$ is a simultaneous minimizer of (almost) all $f_i(x^*)$ [Needell et al. , 2014] . Such a condition would artificially force that the loss function be simultaneously minimized on each training example. In absence of this condition, SGD appears to exhibit similar convergence rate as our paper, albeit under somewhat different assumptions on the loss function.
>
> Moreover, in terms of convergence, we cannot claim that SGMM always outperforms SGM without momentum. For example, in [Kidambi et al. , 2018], the authors show that there exists linear regression problems for which SGM outperforms SGMM in terms of convergence for any learning rate and momentum parameter.

---

> > ### Comment · AnonReviewer2 · 2018-11-27
> > **Response to authors**
> >
> > Dear author(s),
> >
> > Thank you for your response!
> >
> > 1. The set of assumptions (smooth, Lipschitz and strongly convex) is not valid on the whole set R^d,  for example quadratic function. Your L may be arbitrarily large and your bound in (14) could be damaged. I do not think you can properly apply the projection step here after deriving (14) in case of L -> \infty.
> >
> > 2. I was asking about the linear convergence to "neighborhood", not to the "optimal solution". Your theory seems not able to cover this case.

---

> > > ### Author Response · Authors · 2018-11-27
> > > **Response to AnonReviewer2 Comments**
> > >
> > > Dear reviewer,
> > >
> > > - To clarify the correctness of our proof, please note that L in the theorem statement and the proof is bounded due to compactness of the parameter space. We have shown that (19) holds for projected SGMM. Before (19), we have not used L-Lipschitz.
> > >
> > > - Regarding convergence analysis, please note that our goal is to find a *global* convergence bound not a "local" one. We know that classical results show that heavy ball momentum achieves linear convergence rate locally. However, those results are for batch gradient descent not stochastic gradient descent for a general strongly-convex function.

---

> > > > ### Comment · AnonReviewer2 · 2018-11-27
> > > > **Response**
> > > >
> > > > Dear author(s),
> > > >
> > > > You are using a stochastic algorithm. There is nothing guarantee that all of your updated iterations are in compact set. It is true that you consider projected step as in (5). However, your proof in Lemma 1 from the beginning to (14), you are using without projection. It would damage your bound in (14) as I mentioned before that L could be arbitrarily large. Your explanation in the last paragraph of Lemma 1 is not convincing. In order to fix this issue, you may need to consider your derivation with projected step from the beginning of the proof.

---

> > > > > ### Author Response · Authors · 2018-11-28
> > > > > **Response to AnonReviewer2 Comment**
> > > > >
> > > > > Dear reviewer,
> > > > >
> > > > > Please note that inequalities (A.2) and (A.3) (shown in the proof of  Lemma 1 in the supplementary document) hold for the projected SGMM update (5) because Euclidean projection does not increase the distance  between projected points. We are quite certain that our proof is  correct, since our approach to handle projection is a commonly used  technique in existing work. For example, it is used in (Hardt et al., 2016)[Section 3.4].

---

> > > > > > ### Comment · AnonReviewer2 · 2018-11-28
> > > > > > **response**
> > > > > >
> > > > > > I am not sure where I could find the supplementary document as you said. The pdf file only contains 9 pages.

---

> > > > > > > ### Author Response · Authors · 2018-11-28
> > > > > > > **Response to AnonReviewer2 Comment**
> > > > > > >
> > > > > > > Dear reviewer,
> > > > > > >
> > > > > > > The supplementary document was submitted along with the original
> > > > > > > submission. It can be found by clicking the "Show revisions" link below
> > > > > > > the paper title.

---

> > > > > > > > ### Comment · AnonReviewer2 · 2018-11-28
> > > > > > > > **Response**
> > > > > > > >
> > > > > > > > Yes, it should hold for projection steps in (A2) and (A3). But even so, your constant L must depend on the radius of the compact set of the parameters.  How could you guarantee the optimal solution of the strongly convex problem is always in that compact set? In order to increase "the chance" of the compact set containing the optimal solution, you need to "increase" the size of the compact set, which implicitly pushes L be arbitrarily large. Therefore, it is only possible if you just find the solution from the small compact set (from which the optimal solution may be very far).

---

> > > > > > > > > ### Author Response · Authors · 2018-11-29
> > > > > > > > > **Response to AnonReviewer2 Comment**
> > > > > > > > >
> > > > > > > > > Our problem setting involves constrained optimization where we seek the optimal solution within a compact set. This constraint is assumed to be given apriori in the problem definition.  Such setting have been widely considered in the literature. See for example:  (Hardt et al., 2016)[Section 3.4]).
> > > > > > > > >
> > > > > > > > > Please note that we do not address the unconstrained optimization problem that you mention in your response. Thus we do not need to design the compact set that increases the chance of the compact set containing the optimal solution.
> > > > > > > > >
> > > > > > > > > We hope this clarifies our problem setup and you are convinced by the technical soundness of our work.

---

### Official Review · AnonReviewer1 · 2018-11-01

**Rating:** 6
**Confidence:** 4

**Review:**

This paper studies the algorithmic stability of SGD with momentum and provides an upper-bound on true risk through convergence analysis.
This bound clarifies dependencies of convergence speed on the size of dataset and the momentum parameter.

The presentation is easy to follow and technically sounds good.
SGD with momentum is heavily used for learning linear models and deep neural networks, hence to analyze its convergence behavior is quite important.
This paper achieves this goal well by extending a previous result on vanilla SGD in a straightforward manner, although it is not technically difficult.

---

> ### Author Response · Authors · 2018-11-27
> **Response to AnonReviewer1 Comments**
>
> R1C1: Our analysis involves some subtle but important steps in dealing with the momentum term in the recursion in Section 4. This method was not conceived in prior attempts on this problem. To convince you, we reproduce the following statement from [Hardt et al., Section 7]: ``One very important technique that we did not discuss is momentum. However, it is not clear that momentum adds stability. It is possible that momentum speeds up training but adversely impacts generalization.'' Our work is the first successful attempt that establishes that SGMM generalizes, for the practically important class of strongly convex loss functions.

---

> ### Comment · AnonReviewer1 · 2018-11-29
> **Additional comments**
>
> As for the convergence theorem, I think the proof of Theorem 3 for projected SGMM seems correct, but I found another small bug that I did not notice when I read for the first time.
> Proposition 1 cannot be obtained by Theorem 2 and 3 directly because the stability bound is given for the latest parameter $w_t$ while the convergence is guaranteed for the average of ${w_t}$.
> Thus, it would be nice if the authors could fix it.
>
> [Minor typos]
> - "Since \|.\| is a convex function" -> "Since \|.\|^2 is a convex function".
>
> - RHS of Equation after "Note that the LHS of (25) ..." in p. 6:
> 1-\mu -> (1-\mu)^2,
> \| ... \| -> \| ... \|^2.

---

> > ### Author Response · Authors · 2018-11-29
> > **Response to AnonReviewer1 Comments**
> >
> > Dear reviewer,
> >
> > Thank you for your comments and pointing out the typos.
> >
> > Regarding Proposition 1, please note that in our proof (Page 2 in the
> > supplementary document of the original submission, which can be accessed
> > by clicking "show revisions"), we have first shown (as stated in
> > Lemma 2) that the stability bound holds for the average of ${w_t}$.
> > Therefore, we believe that Proposition 1 is correct.

---

> > > ### Comment · AnonReviewer1 · 2018-11-30
> > > **Thank you for your response.**
> > >
> > > I am convinced that proofs are correct and assumptions are also reasonable.

---

### Official Review · AnonReviewer3 · 2018-11-02
**Interesting question and direction**

**Rating:** 4
**Confidence:** 5

**Review:**

This paper presents an analysis of generalization error of SGD with multiple passes for strongly convex objectives using the framework of algorithmic stability [Bousquet and Elisseef, 2002] and its recent use to analyze generalization error of SGD based methods [Hardt, Recht and Singer 2016].

The problem considered by this work is interesting and raises the possibility of understanding generalization related questions of SGD style methods when augmented with momentum, which is common practice in Deep Learning [Sutskever et al. 2013]. That said, there are some concerns about the results as presented in this paper, which I will elaborate below:

- Consider the stability bound admitted by theorem 2: The special case (similar to theorem 3.9 of Hardt et al 2016) when the learning rate alpha = 1/beta (which is the typical learning rate that theory advocates), and setting kappa = beta/gamma where kappa is the condition number of the problem, leads to the following bound on momentum allowed by theorem 2, which is:

(something non-positive) <= mu < 1/(3*kappa).

This is basically the regime where momentum does not make any difference towards accelerating optimization. Referring to the standard value of momentum for strongly convex functions, we see that the momentum is set as mu = (sqrt(kappa)-1)/(sqrt(kappa)+1) [Nesterov, 1983], or, mu = ( (sqrt(kappa)-1)/(sqrt(kappa)+1) )^2  [Polyak,1964]. Upon simplification of this standard momentum values, we see that mu \approx 1 - 1/sqrt(kappa) which grows close to one as kappa grows large. On the other hand, the momentum values admitted by the paper for their bound is super tiny (which gets to zero as the condition number kappa grows large). This essentially implies there is not much about momentum that is captured by the bound of theorem 2 since there is no characterization of the provided bound for theoretically advocated and practically used parameters for momentum.

- In proposition 1, there is no quantitative description of what "sufficiently small" mu (momentum parameter) is - this statement is imprecise. As mentioned in the previous point, sufficiently small mu really is not descriptive of momentum parameters employed in practice (mu in practice typically is >= 0.9). For strongly convex objectives, this should be closer to 1- (1/sqrt(kappa)). Sufficiently small mu parameter essentially does not yield quantitatively different behaviors compared to standard SGD.


In summary, while this paper attempts to make progress on an interesting question, but falls short and doesn't really capture the behavior of these methods that is even mildly reflective of practice (even in terms of the parameter regimes admitted by the bounds proven in the theorems).

- This paper does not perform a thorough literature survey of published results. Furthermore, this paper does not present a precise treatment of assumptions (and implications) amongst other works cited in the paper (see for e.g. [4] below).

[1] Polyak (1987) presents (generalization) behavior of Heavy Ball momentum with noisy (inexact) gradients.
[2] Several efforts in Signal Processing literature do consider the similar setting as one considered by this paper, which is that of Heavy Ball (called accelerated LMS) method with noisy gradients: refer to Proakis (1974), Roy and Shynk (1990), Sharma et al. (1998).
[3] Kidambi et al (2018) estimate the "optimization" power (which is a part of characterization of generalization error [Bach and Moulines 2011], since this dominates at the start of optimization) of HB method with Stochastic Gradients and prove that HB+stochastic gradients does not offer any speedup over vanilla SGD.
[4] Loizou and Richtarik provide an analysis of stochastic heavy ball with super large batch sizes (so they end up showing accelerated rates) under similar assumptions as considered by this paper, such as assuming the function is smooth and strongly convex. However, the paper dismisses the work of Loizou and Richtarik to be working with a different set of assumptions - this is not really true.

---

> ### Author Response · Authors · 2018-11-27
> **Response to AnonReviewer3 Comments**
>
> R3C1: Please note that the theoretically advocated momentum parameters mu = (sqrt(kappa)-1)/(sqrt(kappa)+1) [Nesterov, 1983] or mu = ( (sqrt(kappa)-1)/(sqrt(kappa)+1) )^2 [Polyak,1964] are based on the *convergence* analysis of GD with momentum -- they do not account for *generalization*. Therefore, these values are not necessarily optimal for SGD with momentum (SGMM), in terms of our objective of true risk. In Theorem 2, our focus is to find a bound on stability, i.e., the condition on generalization. Our theorem for convergence analysis (Theorem 3) does not have any limitation on mu.
>
> Our goal in Theorem 2 is to find the tightest possible bound that shows why machine learning models can be trained for multiple epochs of SGMM while their generalization errors are limited. In order to satisfy uniform stability for SGMM (with constant momentum), we need to have a recursion with coeff<1. Even ignoring the third term in the RHS of (12), we still have to assume mu<1/kappa in order to have such a recursion.
>
> We agree theoretically suggested momentum parameters for GD approach 1 – 1/sqrt(kappa), which grow close to one as kappa grows large. On the other hand, as our concern in this work is on gamma-strongly convex loss functions, where the gamma parameter can be tuned by adjusting a weight decay regularization parameter in a typical machine learning model, it is indeed important and interesting to study the generalization bound when kappa is not too large, i.e., the non-asymptotic regime. When kappa is not too large, the range specified in our theorem captures a typical value of the suggested momentum based on convergence analysis of GD. As an example, if we set kappa = 3.5, then ((sqrt(kappa)-1)/(sqrt(kappa)+1))^2≈0.1, which is approximately 1/(3*kappa).
> ------------------------------------------------------------------------------------------------------------------------------------------------------------------------
> R3C2: In the original submission, we specified the condition mu*(beta+gamma)<<alpha*beta*gamma in the supplementary document. In the revision, we have explicitly provided this condition it in the proposition statement. Please note that this condition is used only to make tractable the optimization of the expected true risk over alpha. In practice, we can still use alpha as specified in Proposition 1. However, it will not necessarily optimize the upper-bound on the expected true risk.
> ------------------------------------------------------------------------------------------------------------------------------------------------------------------------
> R3C3: Please note that although [1]-[3] study first-order methods with noisy (imperfect) gradients, none of these works study generalization of SGD for a strongly convex loss function using algorithmic stability. We note that both [1] and [2] are cited in [3]. In the revision, we have added the following sentence to our introduction: "First-order methods with noisy gradient are studied in [Kidambi et al. , 2018] and references therein. In [Kidambi et al. , 2018], the authors show that there exists linear regression problems for which SGM outperforms SGMM in terms of convergence."
>
> Regarding comparison with [Loizou et al. , 2018], please note that [Loizou et al. , 2018] considers the special case of a convex *quadratic* loss function of a least-squares type, while we consider the general case of strongly convex loss functions. Furthermore, we emphasize that we do not limit our analysis of SGMM to super large batch sizes. Our analysis indeed works even for a batch size of one.

---

> > ### Comment · AnonReviewer3 · 2018-11-29
> > **response**
> >
> > Thank you for your response.
> >
> > Firstly, please note that in Kidambi et al (2018), the paper presents the fact that SGD + momentum (for any value of the learning rate + momentum tuple) does not outperform vanilla SGD (with mu=0) by more than a constant factor (i.e. there is no asymptotic running time improvements in the big-Omega/big-Oh notation). That said, momentum + SGD can never be worse than vanilla SGD since we can always set mu=0 in momentum methods to recreate SGD's behavior.
> >
> > With regards to values of momentum for generalization: There are generalization results for acceleration with stochastic gradient methods. In particular, Jain et al. "accelerating stochastic gradient descent for least squares regression" 2017 present (for the strongly convex least squares problem) a result that admits a similar flavor - where, momentum approaches 1 for harder problems (large condition number) than ones for easier problems (low condition number).
> >
> > At the end of the day, the trend offered by both deterministic and/or stochastic accelerated methods for easy (low condition number) vs. hard (large condition number) problems is what matters: for harder problems with large condition number, the momentum approaches 1, but, for the bounds admitted by the paper, the momentum approaches zero, which is rather worrying. This basically implies that this paper's theory, while beginning to make progress on this problem, does not provide a reasonable guarantee to characterize what happens when SGD is used with momentum.
> >
> > And a condition number of 3.5? A typical (easy) machine learning problem has a condition number around O(10^3) or more. Harder (and more typical) ones that have many correlated features have a condition number that is roughly 10^5 or more with greater correlations across features leading to worsening of the condition number. A condition number of 3.5 is trivial (from an optimization standpoint) - gradient descent requires roughly 10-20 steps to converge on this problem. A condition number of 3.5 implies that there is close to little advantage of kappa versus sqrt(kappa), which is the advantage offered by acceleration. Finally, in order to make the condition number 3.5, one would have to regularize the problem too strongly in a way that the solution will very well generalize far worse than solving the erm problem.
> >
> > Again, I don't understand the claim of why a smaller kappa resembles "non-asymptotics". Precisely put, there is *no* relation of kappa versus asymptotics.

---

> > > ### Author Response · Authors · 2018-12-05
> > > **Response to AnonReviewer3 Comments**
> > >
> > > Dear reviewer,
> > >
> > > Regarding the comparison with (Jain et al., 2018), we should clarify that only linear regression problem with quadratic loss function has been studied in (Jain et al., 2018), while we consider a general strongly-convex loss function. In addition, our generalization bound is based on uniform stability, which is not the case in (Jain et al., 2018). Hence, we do not find a strong connection between the results of (Jain et al., 2018) and those of our paper.
> > >
> > > We emphasize that the claim that "for problems with larger condition number, the momentum should approach to one" is indeed based on *convergence* analysis of GD with momentum. It does not account for *generalization*. In addition, those recommended parameters are not necessarily optimal for SGD. In our view, what mainly matters is extending the results of (Hardt et al., 2016) to SGMM and showing that there exists some mu for which SGMM satisfies uniform stability, i.e., our machine learning model can be trained for multiple epochs and still generalizes. We have verified the trends predicted by our stability bounds using experimental evaluations. Those evaluations are based on common machine learning models leading to a smooth and strongly convex loss function.
> > >
> > > By asymptotic, we mean that the problem structure imposes very large *kappa*. As explained before, the gamma parameter can be tuned by adjusting a weight decay regularization parameter in a typical machine learning model. We used 3.5 merely as an example to show that there exists mathematical problems for which the suggested momentum based on the convergence analysis of GD falls within the interval specified by Theorem 2 in our paper. We do not claim that kappa=3.5 necessarily represents practical problems in machine learning. Please note that even for the original work of  (Hardt et al., 2016), which analyzes the stability of SGD without momentum, there are some conditions on the learning rate in the theorem statements to satisfy the uniform stability, i.e., unlike the convergence analysis, the stability bounds typically involve limitations on the range of hyper-parameters. We further emphasize that our theorem for convergence analysis (Theorem 3) does not have any limitation on mu.

---

### Public Comment · (anonymous) · 2019-03-27
**carmodel**

Collectible cars have always been and remain an excellent gift for both a child and an adult. For example, if your friend dreams of a particular make or model of car, or perhaps he has one, then why not give him a scale model of his favorite car?

Many collectible models https://www.bestadvisor.com/car-model-kits have a metal case and even prefab models are no exception. Most models open doors, luggage compartment, hood, and even turns the steering wheel in the cabin. All these possibilities depend on the specific model of the souvenir machine and its features. Large-scale collectibles such as 1:18 or 1:24 always have a lot of extra features. And for 1:43 or 1:34 scale models, only the doors are usually opened.

---

### Meta-Review · Area_Chair1 · 2018-12-12
**Paper needs more work**

**Confidence:** 5
**Recommendation:** Reject

**Metareview:**

The paper according to Reviewers needs more work for publication and significantly more clarifications. The Reviewers are not convinced on publishing even after intensive discussion that the AC read in full. The AC recommends further improvements on the paper to address better Reviewer's concerns.